# Caspase-1 interdomain linker cleavage is required for pyroptosis

Daniel P Ball[1,*], Cornelius Y Taabazuing[1,*], Andrew R Griswold[2], Elizabeth L Orth[3], Sahana D Rao[3], Ilana B Kotliar[3], Lauren E Vostal[3], Darren C Johnson[3], Daniel A Bachovchin[1,3,4]

**Pathogen-related signals induce a number of cytosolic pattern-recognition receptors (PRRs) to form canonical inflammasomes, which activate pro-caspase-1 and trigger pyroptotic cell death. All well-studied inflammasome-forming PRRs oligomerize with the adapter protein ASC (apoptosis-associated speck-like protein containing a CARD) to generate a large structure in the cytosol, which induces the dimerization, autoproteolysis, and activation of the pro-caspase-1 zymogen. However, several PRRs can also directly interact with pro-caspase-1 without ASC, forming smaller "ASC-independent" inflammasomes. It is currently thought that little, if any, pro-caspase-1 autoproteolysis occurs during, and is not required for, ASC-independent inflammasome signaling. Here, we show that the related human PRRs NLRP1 and CARD8 exclusively form ASC-dependent and ASC-independent inflammasomes, respectively, identifying CARD8 as the first canonical inflammasome-forming PRR that does not form an ASC-containing signaling platform. Despite their different structures, we discovered that both the NLRP1 and CARD8 inflammasomes require pro-caspase-1 autoproteolysis between the small and large catalytic subunits to induce pyroptosis. Thus, pro-caspase-1 self-cleavage is a required regulatory step for pyroptosis induced by human canonical inflammasomes.**

## Introduction

Caspase-1 is a cysteine protease that induces pyroptotic cell death in response to a number of pathogen-associated signals (Lamkanfi & Dixit, 2014; Broz & Dixit, 2016). Typically, an intracellular pattern recognition receptor (PRR) detects a particular microbial structure or activity, self-oligomerizes, and recruits the adapter protein ASC (apoptosis-associated speck-like protein containing a CARD), which in turn polymerizes to form an "ASC focus" in the cytosol (Yu et al, 2006; Jones et al, 2010; Broz et al, 2010a). Full-length caspase-1 zymogen (or pro-caspase-1) is recruited to this structure, where it is

activated by proximity-induced autoproteolysis. Active caspase-1 then cleaves and activates the inflammatory cytokines pro-IL-1β and pro-IL-18 and the pore-forming protein gasdermin D (GSDMD), causing inflammatory cell death (Kayagaki et al, 2015; Shi et al, 2015). Collectively, the structures that activate pro-caspase-1 are called "canonical inflammasomes."

Two death-fold domains, the pyrin domain (PYD) and the caspase activation and recruitment domain (CARD), mediate canonical inflammasome assembly (Broz & Dixit, 2016). ASC is comprised of a PYD and a CARD (Fig 1A) and bridges either the PYD or CARD of an activated PRR to the CARD of pro-caspase-1 via homotypic interactions. In mice, all known pro-caspase-1–activating PRRs form ASC-containing inflammasomes. However, in the absence of ASC, two murine CARD-containing PRRs, NLRC4 and NLRP1B, can directly recruit and activate pro-caspase-1 through CARD–CARD interactions (Poyet et al, 2001; Mariathasan et al, 2004; Broz et al, 2010b; Guey et al, 2014; Van Opdenbosch et al, 2014). ASC-independent inflammasomes induce the cleavage of GSDMD and trigger lytic cell death, but do not form foci or efficiently process pro-caspase-1 and pro-IL-1β (Broz et al, 2010b; He et al, 2015).

These observations indicated that pro-caspase-1 autoproteolysis may not be required for the induction of cell death. To explore this possibility, two independent groups reconstituted $Casp1^{-/-}$ mouse macrophages with an uncleavable mutant form of mouse pro-caspase-1 and found that the mutant enzyme still mediated cell death but did not process pro-IL-1β in response to various inflammasome stimuli (Broz et al, 2010b; Guey et al, 2014). Another study, performed after the discovery of GSDMD, showed that this uncleavable mutant pro-caspase-1 was at least partially defective in processing GSDMD and inducing pyroptosis in RAW 264.7 cells in response to NLRP3 inflammasome activation (He et al, 2015). Regardless, these studies indicated that mouse pro-caspase-1 self-processing is not required for pyroptosis and that ASC-independent inflammasomes specifically activate pro-caspase-1 without inducing much autoproteolysis. The requirement of human pro-caspase-1 autoproteolysis for the induction of pyroptosis has not been evaluated experimentally.

DPP8/9 inhibitors activate the human NLRP1 and CARD8 inflammasomes (Fig 1A), which both have C-terminal ZU5 (found in ZO-1 and

[1]Chemical Biology Program, Memorial Sloan Kettering Cancer Center, New York, NY, USA   [2]Weill Cornell/Rockefeller/Sloan Kettering Tri-Institutional MD-PhD Program, New York, NY, USA   [3]Tri-Institutional PhD Program in Chemical Biology, Memorial Sloan Kettering Cancer Center, New York, NY, USA   [4]Pharmacology Program of the Weill Cornell Graduate School of Medical Sciences, Memorial Sloan Kettering Cancer Center, New York, NY, USA

Correspondence: bachovcd@mskcc.org
*Daniel P Ball and Cornelius Y Taabazuing contributed equally to this work

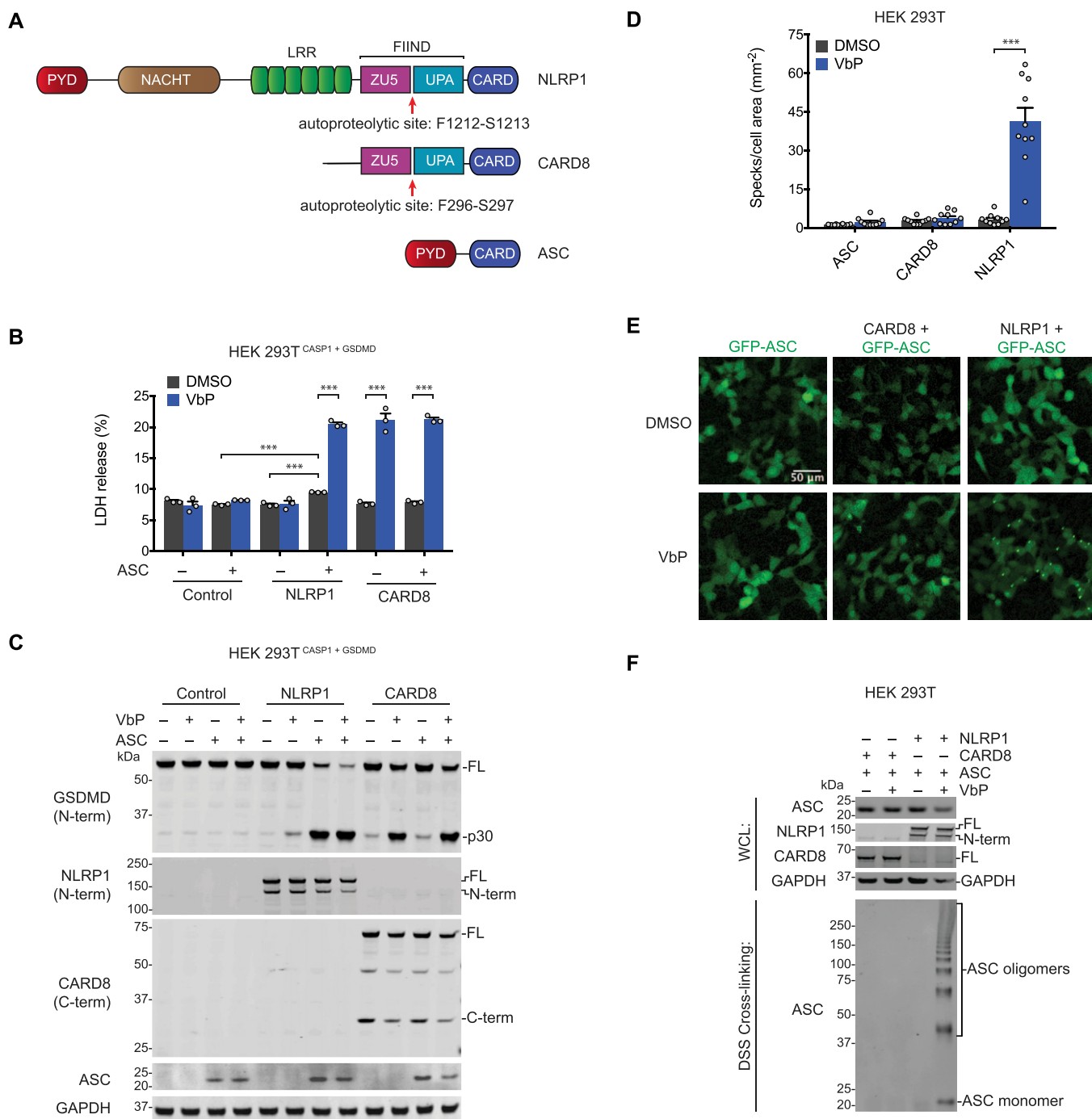

**Figure 1. NLRP1 is ASC-dependent and CARD8 is ASC-independent.**
**(A)** Human NLRP1, CARD8, and ASC domain organization. The autoproteolysis sites are indicated. **(B, C)** HEK 293T cells stably expressing CASP1 and GSDMD (HEK 293T[CASP1 + GSDMD]) were transiently transfected with constructs encoding the indicated proteins and treated with DMSO or VbP (10 μM, 6 h). **(B, C)** Supernatants were evaluated for LDH release (B) and lysates were analyzed by immunoblotting (C). Data are means ± SEM of three biological replicates. ***P < 0.001 by two-sided t test. **(D, E)** HEK 293T cells were transfected with constructs encoding GFP-tagged ASC and NLRP1 or CARD8, treated with DMSO or VbP (10 μM, 6 h), and evaluated for ASC speck formation by fluorescence microscopy. The cells were not fixed before analysis. **(D, E)** Shown are the mean ± SEM (D) and representative images (E) from 10 replicates from one of two independent experiments. ***P < 0.001 by two-sided t test. **(F)** HEK 293T cells transiently transfected with constructs encoding the indicated proteins and treated with DMSO or VbP (10 μM, 6 h). Lysates were harvested, subjected to disuccinimidyl suberate cross-linking, and evaluated by immunoblotting. All data, including immunoblots, are representative of three or more independent experiments. FL, full-length; WCL, whole cell lysate.

UNC5), UPA (conserved in UNC5, PIDD, and ankyrin), and CARD domains (Okondo et al, 2017; Johnson et al, 2018; Zhong et al, 2018; Gai et al, 2019). The ZU5 domains of NLRP1 and CARD8 undergo posttranslational autoproteolysis (Fig 1A), generating non-covalently associated, auto-inhibited N- and C-terminal polypeptide fragments (D'Osualdo et al, 2011; Finger et al, 2012; Frew et al, 2012). The C-terminal UPA-CARD fragments mediate cell death after the autoinhibitory N terminus is degraded by the proteasome (Finger et al, 2012; Johnson et al, 2018; Chui et al, 2019; Sandstrom et al, 2019). CARD8 does not require ASC to activate pro-caspase-1 (Okondo et al, 2017; Johnson et al, 2018), but it is unknown whether CARD8 can also form an ASC-containing inflam-masome. In contrast, human NLRP1, unlike mouse NLRP1A and NLRP1B (Masters et al, 2012; Van Opdenbosch et al, 2014), appears to require ASC to activate pro-caspase-1 (Finger et al, 2012; Zhong et al, 2016, 2018).

Here, we show that CARD8 and NLRP1 exclusively form ASC-independent and ASC-dependent inflammasomes, respectively, due to specific CARD–CARD interactions. These data identify CARD8 as the first pro-caspase-1–activating PRR that does not form an ASC focus. The CARD8 inflammasome, like the mouse ASC-independent inflammasomes, induces little detectable pro-caspase-1 process-ing by immunoblotting (Okondo et al, 2017; Johnson et al, 2018). Surprisingly, however, we found that both the NLRP1 and CARD8 inflammasomes require human pro-caspase-1 autoproteolysis to induce GSDMD cleavage and pyroptosis. Moreover, we discovered that a mutation (D308N) in the uncleavable mouse construct dysre-gulates the enzyme to induce GSDMD-independent cell death, and that mouse pro-caspase-1 self-processing is similarly required for pyrop-tosis. Overall, these data demonstrate that caspase-1 autoproteolysis is critical for canonical inflammasome signaling.

## Results and Discussion

### NLRP1 is ASC-dependent and CARD8 is ASC-independent

We first wanted to determine the capabilities of human NLRP1 and CARD8 to form ASC-dependent and ASC-independent inflamma-somes. Therefore, we transfected constructs encoding NLRP1, CARD8, and/or ASC into HEK 293T cells stably expressing pro-caspase-1 and GSDMD before treatment with the DPP8/9 inhibitor Val-boroPro (VbP). VbP induced similar levels of GSDMD cleavage and lactate dehydrogenase (LDH) release in cells expressing CARD8 in the presence or absence of ASC (Fig 1B and C), confirming that ASC is not required for CARD8-mediated cell death (Okondo et al, 2017; Johnson et al, 2018). In contrast, NLRP1 required ASC co-expression to mediate cell death (Fig 1B and C). We should note that the co-expression of NLRP1 and ASC induced some spontaneous cell death and GSDMD cleavage, but both were increased by VbP. Consistent with these data, transient transfection of constructs encoding the active UPA-CARD fragment of NLRP1, but not CARD8, required ASC to induce GSDMD cleavage (Fig S1A). As previously reported, the PYD of NLRP1 was dispensable for inflammasome activation (Fig S1B and C) (Finger et al, 2012; Chavarría-Smith et al, 2016).

Although these results confirm that CARD8 can directly activate pro-caspase-1 without ASC bridging, it remained possible that CARD8 could also form an ASC-containing inflammasome, similar to

mouse NLRP1B (Van Opdenbosch et al, 2014; Gai et al, 2019). We next co-transfected HEK 293T cells with constructs encoding GFP-tagged ASC and either NLRP1 or CARD8. These cells were then treated with VbP for 6 h and imaged by fluorescence microscopy (Fig 1D and E). VbP induced ASC specks in NLRP1, but not CARD8, expressing cells, suggesting that CARD8 cannot form an ASC speck-containing inflammasome. Similarly, transfection of the UPA-CARD of NLRP1, but not CARD8, induced ASC speck formation (Fig S1D and E). To further support these microscopy results, we co-transfected HEK 293T cells with constructs encoding untagged ASC and either NLRP1 or CARD8, treated the cells with VbP, and cross-linked lysates with disuccinimidyl suberate (DSS). As expected, VbP induced ASC oligomerization in cells expressing NLRP1, but not CARD8 (Fig 1F).

We hypothesized that the exclusive formation of ASC-independent and ASC-dependent inflammasomes by CARD8 and NLRP1, respectively, was due to specific interaction differences between the CARDs of CARD8 and NLRP1 with the CARDs of ASC and CASP1. To test this prediction, we incorporated these CARDs into the split luciferase-based NanoBiT assay (Dixon et al, 2016), fusing Small BiT (SmBiT, an 11 amino acid peptide) to the CARD domains of ASC and CASP1 and Large BiT (LgBiT, an 18-kD tag that forms a functional luciferase enzyme when bound to the SmBiT peptide) to the CARD domains of ASC, CASP1, CARD8, and NLRP1 (Fig 2A). We mixed lysates containing the indicated fusion proteins and ob-served luminescent signals indicating binding between the ASC$^{CARD}$ and itself, CASP1$^{CARD}$, and NLRP1$^{CARD}$ (Fig 2B) and between the CASP1$^{CARD}$ and itself, ASC$^{CARD}$, and CARD8$^{CARD}$ (Fig 2C). As expected, we did not observe a CASP1$^{CARD}$–NLRP1$^{CARD}$ interaction or an ASC$^{CARD}$–CARD8$^{CARD}$ interaction. Similarly, we observed that full-length ASC interacted with NLRP1$^{CARD}$, but not CARD8$^{CARD}$ (Fig S1F). Overall, these results indicate specific CARD–CARD interactions govern the formation of the CARD8 ASC-independent inflamma-some and the NLRP1 ASC-dependent inflammasome.

### The CARD8 inflammasome requires caspase-1 processing

We initially discovered DPP8/9 inhibitor-induced pyroptosis in human THP-1 cells (Okondo et al, 2017), which is mediated by CARD8 (Johnson et al, 2018). We observed little, if any, caspase-1 and IL-1β processing, and thus designated this death as "pro-caspase-1-dependent" pyroptosis. However, we never formally demonstrated that pro-caspase-1 itself mediates this response. We next wanted to determine if the CARD8 ASC-independent inflammasome can in-duce pro-caspase-1 processing. VbP stimulates a slower, apoptotic form of cell death in $GSDMD^{-/-}$ cells, which eventually release intracellular contents via secondary necrosis (Taabazuing et al, 2017; Tsuchiya et al, 2019). We reasoned that pro-caspase-1 pro-cessing might be observed in $GSDMD^{-/-}$ THP-1 cells, as more self-cleavage might occur over the longer time interval and the cleavage products might not be as readily released into the supernatant. As expected, VbP induced LDH release (Fig 3A) and poly (ADP-ribose) polymerase (PARP) cleavage (Fig 3B) in $GSDMD^{-/-}$ THP-1 cells. Consistent with our hypothesis, we observed bands corresponding to the p10 and p20 fragments of caspase-1 in the lysates from VbP-treated $GSDMD^{-/-}$ cells, but not from sg$GFP$ control cells (Fig 3B). Moreover, we also observed a faint band corresponding to the p20 in concentrated supernatants from VbP-treated sg$GFP$ cells,

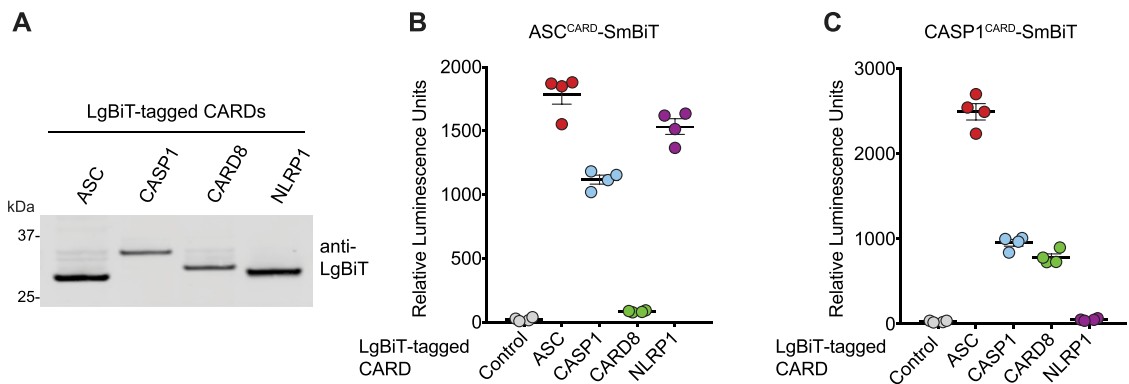

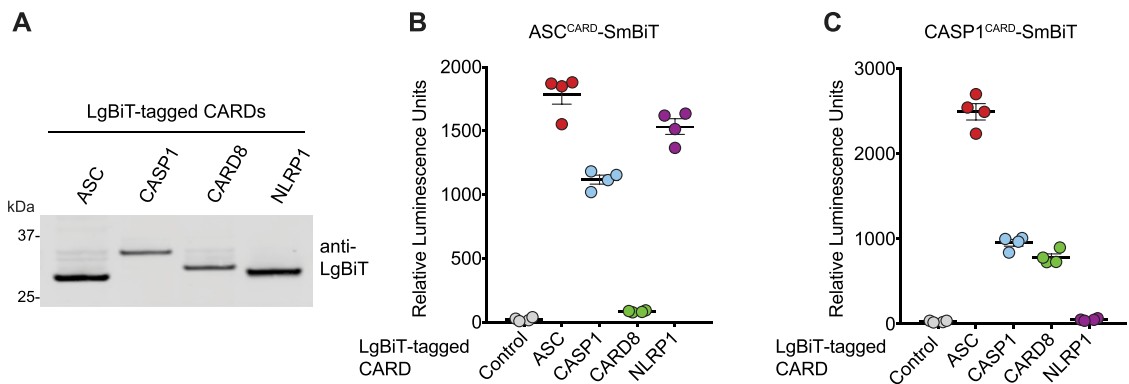

**Figure 2. Specific CARD–CARD interactions determine ASC-dependent or independent inflammasome assembly.**
**(A)** Expression of the indicated LgBiT-tagged CARDs in HEK 293T cells was verified by immunoblotting. **(B, C)** Cell lysates from HEK 293T cells transiently expressing SmBiT-tagged ASC^CARD (B) or SmBiT-tagged CASP1^CARD (C) were mixed with lysates expressing LgBiT-tagged CARDs and analyzed for the relative luminescence. Data are means ± SEM of four independent replicates.

consistent with its release during pyroptosis. These results indicate that the CARD8 inflammasome can, in fact, process pro-caspase-1.

We next wanted to determine if pro-caspase-1 processing was required for cell death. Analogous to the previously created uncleavable mouse pro-caspase-1 (mCASP1 D6N) (Broz et al, 2010b), we generated an uncleavable human pro-caspase-1 (CASP1 D5N, Fig 3C) in which all Asp cleavage sites were mutated to Asn residues (Thornberry et al, 1992). We generated HEK 293T cell lines stably expressing GSDMD and wild-type (WT), uncleavable (D5N), or catalytically inactive (C285A) pro-caspase-1, and then transiently transfected constructs encoding WT or autoproteolytic-defective (inactive) S297A CARD8 into each of these cell lines. As expected, VbP induced cell death and GSDMD cleavage in cells with WT pro-caspase-1 and WT CARD8, but not in cells expressing catalytically dead CASP1 or autoproteolysis-defective CARD8 (Fig 3D). We also observed a small amount of the p20 cleaved product in the cell line expressing CASP1 WT (Fig 3E). Interestingly, we did not observe cell death or GSDMD cleavage in cells expressing the uncleavable CASP1 D5N (Fig 3D and E), indicating that pro-caspase-1 autoproteolysis is needed for CARD8 inflammasome-induced pyroptosis.

### Human pro-caspase-1 interdomain linker (IDL) cleavage is required for pyroptosis

We next wanted to determine which specific pro-caspase-1 cleavage events were required for CARD8-mediated death. Pro-caspase-1 comprises three domains, a CARD, a large subunit (p20), and a small subunit (p10), separated by two linkers (Fig 3C). Pro-caspase-1 undergoes proteolytic processing at two sites (D103 and D119) in the CARD linker (CDL) that separates the CARD and the p20 and at three sites (D297, D315, and D316) in the IDL that separates the p20 and the p10 (Thornberry et al, 1992; Boucher et al, 2018). As IDL cleavage has been associated with higher catalytic activity and CDL cleavage with termination of activity (Elliott et al, 2009; Broz et al, 2010b; Boucher et al, 2018), we first evaluated the importance of the three cleavage sites in the IDL by generating CASP1 D297N, D315N/D316N, and D297N/D315N/D316N ("IDL uncleavable," or IDL^uncl) mutant constructs. We then transiently transfected these constructs with and without a construct encoding the UPA-CARD

fragment of CARD8 into HEK 293T cells stably expressing GSDMD. As expected, the CARD8^UPA-CARD robustly induced the formation of the p30 fragment of GSDMD in cells expressing WT pro-caspase-1 (Fig 4A). In contrast, we observed markedly reduced p30 fragment in cells expressing the D297N or D315N/D316N mutants, and none in cells expressing the IDL^uncl or catalytically dead C285A enzymes (Fig 4A). We did not observe any detectable pro-caspase-1 processing in this experiment, similar to the results with the THP-1 cells expressing GSDMD (Fig 3B).

To more easily observe pro-caspase-1 processing in lysates, we next generated HEK 293T cells stably expressing the pro-caspase-1 constructs in the absence of GSDMD. We then transfected these cells with either CARD8^UPA-CARD or NLRP1^UPA-CARD plus ASC. As expected, we observed the p20, p22, and p10 caspase-1 fragments in cells expressing WT, D297N, or D315N/D316N constructs (Fig 4B), which are capable of undergoing IDL processing. In addition, we observed processing of the IDL^uncl caspase-1 into a p35 fragment, consistent with CDL cleavage and no processing at all of the C285A or D5N constructs (Fig 4B). Similar pro-caspase-1 and GSDMD processing was observed after transfecting full-length CARD8 or full-length NLRP1 plus ASC and treating the cells with VbP (Fig S2A and B), as well as after transfecting a construct encoding residues 1–328 of human NLRC4, which contains a CARD domain that directly activates human CASP1 independent of ASC (Poyet et al, 2001) (Fig S2C). Also as expected, the CARD8 inflammasome-induced PARP cleavage only in cells lacking GSDMD and expressing the WT, D297N, or D315N/D316N constructs (Figs 4B and S2A). In contrast, the active NLRP1 inflammasome-induced PARP cleavage in all cells lacking GSDMD (Figs 4B and S2A), consistent with the well-established ability of ASC-containing inflammasomes to activate caspase-8 independent of caspase-1 (Pierini et al, 2012; Sagulenko et al, 2013; Van Opdenbosch et al, 2017).

Replacing CARDs with the DmrB domain enables the small-molecule (AP-20187)–induced dimerization, autoproteolysis, and activation of caspases (Oberst et al, 2010; Boucher et al, 2018; Ross et al, 2018; Ruhl et al, 2018). To confirm that IDL cleavage was required for proximity-induced pro-caspase-1 activation, we cloned DmrB-caspase-1 constructs with the IDL mutations described above (Fig S2D). We transiently transfected these constructs into HEK 293T

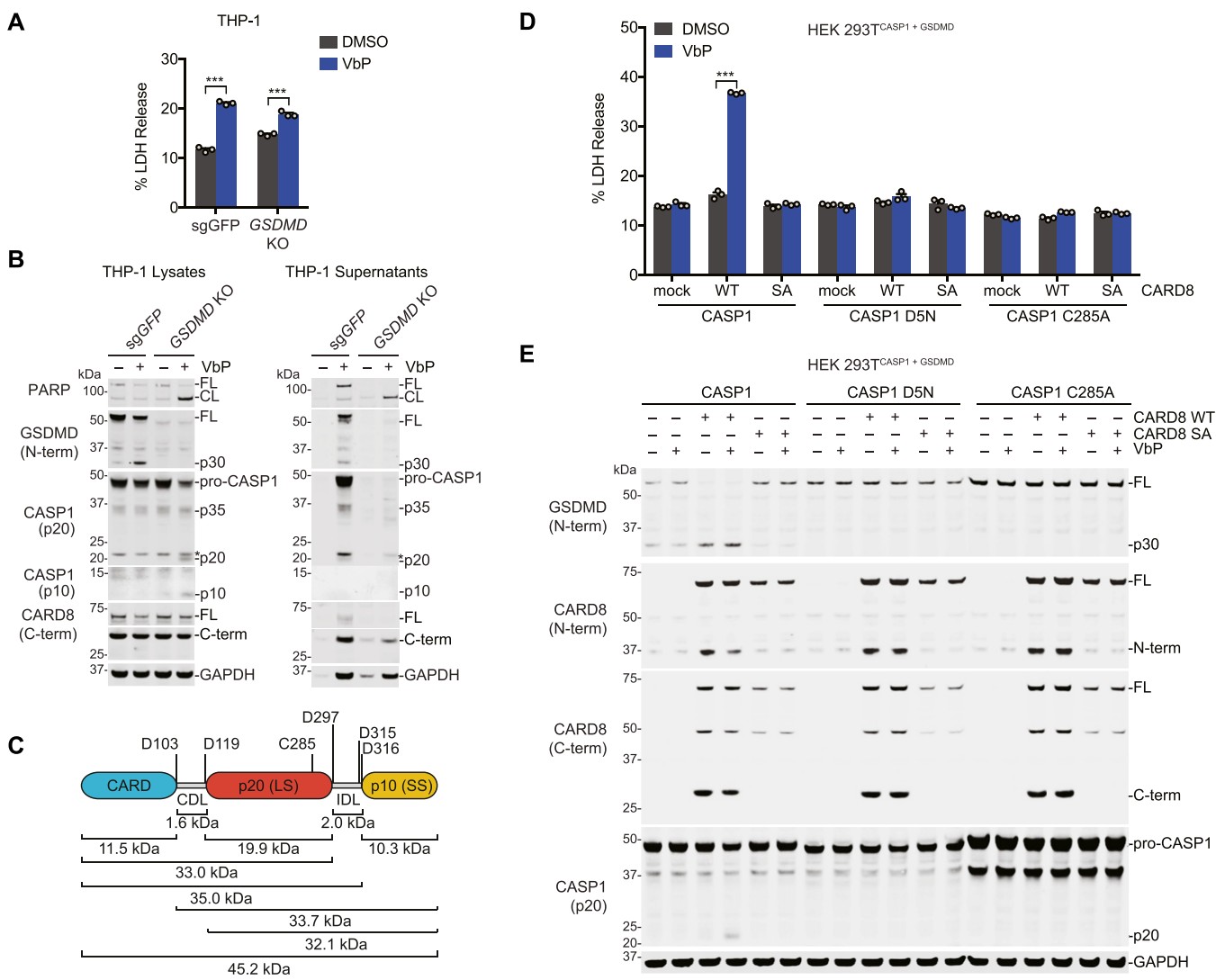

**Figure 3. Caspase-1 autoproteolysis is required for CARD8-mediated death.**
**(A, B)** Control and *GSDMD*$^{-/-}$ THP-1 cells were treated with VbP (10 $\mu$M, 24 h) before supernatants were analyzed for LDH release (A) and lysates and supernatants were evaluated by immunoblotting (B). Data are means ± SEM of three biological replicates. ***$P < 0.001$ by two-sided $t$ test. An asterisk indicates a background band. **(C)** Schematic of pro-caspase-1 depicting the CARD domain and large (p20, LS) and small (p10, SS) catalytic subunits. Predicted cleavage sites, sizes of potential cleavage products, and the catalytic cysteine are indicated. **(D, E)** HEK 293T cells stably expressing GSDMD and the indicated pro-caspase-1 constructs were transiently transfected with plasmids encoding RFP (mock), CARD8 WT, or autoproteolysis-defective CARD8 S297A (SA) for 24 h before addition of VbP (10 $\mu$M, 6 h). **(D, E)** Cell death was assessed by LDH release (D) and GSDMD and CASP1 cleavage by immunoblotting (E). Data are means ± SEM of three biological replicates. ***$P < 0.001$ by two-sided $t$ test. All data, including immunoblots, are representative of three or more independent experiments. CL, cleaved; FL, full-length.

cells, and then treated with AP-20187. We observed that the WT DmrB-caspase-1 underwent significant autoproteolysis and triggered GSDMD cleavage (Fig S2E). Some pro-caspase-1 autoproteolysis and GSDMD cleavage were also observed for D297N and the D315N/D316N mutants, but not the IDL$^{uncl}$ mutant.

We next wanted to confirm the importance of pro-caspase-1 processing for the induction of pyroptosis in monocytic cells. Thus, we ectopically expressed GFP or pro-caspase-1 mutants in *CASP1*$^{-/-}$ THP-1 cells (Fig 4C). As expected, VbP induced GSDMD processing and LDH release in control (sg*GFP*-treated), but not in *CASP1*$^{-/-}$ THP-1 cells (Fig 4C and D). Moreover, VbP induced GSDMD processing in *CASP1*$^{-/-}$ THP-1 cells reconstituted with WT pro-caspase-1. Consistent with the results in HEK 293T cells (Fig 4A),

GSDMD cleavage was considerably reduced in cells expressing D297N or D315N/D316N pro-caspase-1 and absent in cells expressing the IDL$^{uncl}$, C285A, or D5N pro-caspase-1 constructs (Fig 4C). Overall, these data strongly indicate that caspase-1 IDL processing is required for GSDMD cleavage and pyroptosis. However, we should note that the mutant proteins used to derive this conclusion could, in theory, also be defective in ways unrelated to autoproteolysis.

Intriguingly, we observed PARP cleavage (Fig 4C) and LDH release (Fig 4D) in *CASP1*$^{-/-}$ THP-1 cells ectopically expressing IDL$^{uncl}$, D5N, or C285A pro-caspase-1 constructs. Pro-caspase-1 can have scaffolding roles in addition to its catalytic function (Van Opdenbosch et al, 2017), and we hypothesized that these

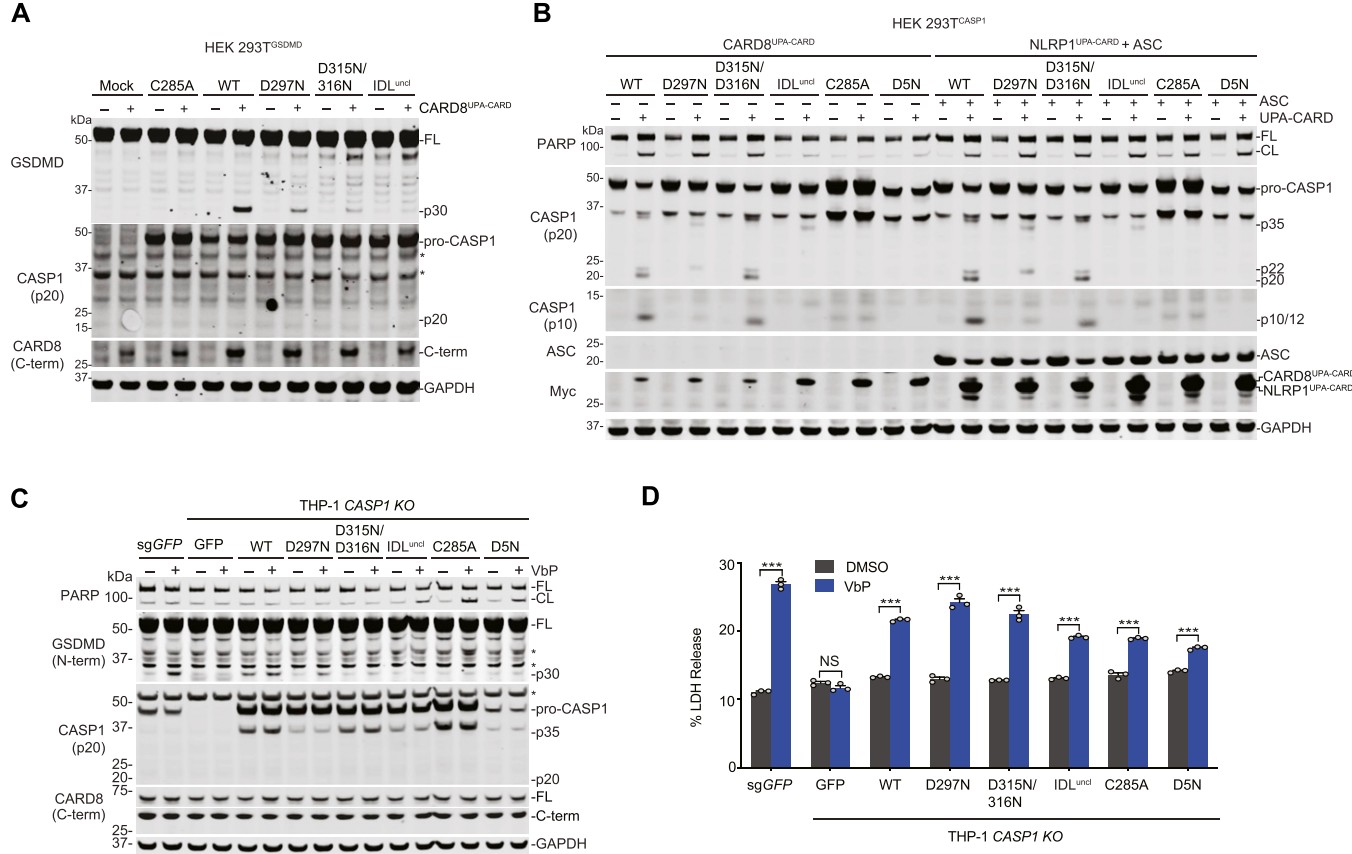

**Figure 4. Human pro-caspase-1 interdomain linker cleavage is required for pyroptosis.**
**(A)** HEK 293T cells stably expressing GSDMD were transiently transfected with plasmids encoding RFP (mock), the indicated pro-caspase-1 constructs, and the CARD8UPA-CARD for 24 h before lysates were analyzed by immunoblotting. Asterisks indicate background bands. **(B)** HEK 293T cells stably expressing the indicated pro-caspase-1 constructs were transiently transfected with plasmids encoding CARD8UPA-CARD or NLRP1UPA-CARD and ASC for 24 h before lysates were evaluated by immunoblotting. **(C, D)** Control and *CASP1−/−* THP-1 cells ectopically expressing the indicated pro-caspase-1 proteins were treated with DMSO or VbP (10 μM, 24 h). **(C, D)** GSDMD cleavage was assessed by immunoblotting (C) and cell death by LDH release (D). Data are means ± SEM of three biological replicates. ***$P < 0.001$ by two-sided $t$ test. An asterisk indicates background bands. All data, including immunoblots, are representative of three or more independent experiments. CL, cleaved; FL, full-length.

defective pro-caspase-1 mutants were bridging the CARD8UPA-CARD to ASC and thereby activating apoptosis. Indeed, the co-expression of ASC and CARD8UPA-CARD induced PARP cleavage in HEK 293T cells only in the presence of C285A mutant pro-caspase-1 (Fig S3A). In contrast and as expected, the expression of NLRP1UPA-CARD and ASC in HEK 293T cells, which triggers ASC speck formation (Fig S1D and E), was sufficient to induce PARP cleavage without expression of C285A pro-caspase-1 (Fig S3B).

## Cleavage of the mouse pro-caspase-1 IDL is critical for pyroptosis

Unlike human pro-caspase-1, mouse pro-caspase-1 has an Asp residue (D308) within its IDL linker that can be cleaved during autoproteolysis (Fig 5A) (Broz et al, 2010b). As such, Broz and co-workers mutated this site, as well as D103, D122, D296, D313, and D314, to Asn to create the uncleavable mouse D6N protein. We next wanted to investigate if mouse D6N, unlike mouse C284A or human D5N, was indeed capable of mediating pyroptosis. Thus, we transiently transfected constructs encoding wild-type, D6N, D3N (D296N/D313N/D314N), or D4N (D296N/D308N/D313N/D314N) mouse pro-caspase-1 with and without NLRP1BUPA-CARD into HEK 293T cells stably expressing mouse GSDMD (Fig 5B–E). Intriguingly, we observed

that both the D6N and D4N mutants, but not the WT or D3N enzymes, induced cell death even in the absence of NLRP1BUPA-CARD (Fig 5B and D), indicating that the D308N mutation impaired enzyme auto-inhibition. The presence of an Asp at residue 308 appears to be critical, as mutation to either Ala or Glu (D3N/D308A or E) also dys-regulated basal enzyme activity (Fig 5D). Despite their ability to trigger cell death, all IDL mutant enzymes, including the D3N mutant, were severely defective at cleaving GSDMD (Fig 5C and E). We next generated HEK 293T cells stably expressing these mouse pro-caspase-1 constructs to more easily visualize IDL processing. The WT and D3N enzymes, but not the D3N/D308A or E mutants, underwent IDL processing and cleaved GSDMD (Fig 5F and G). However, the D3N mutant was again severely impaired in cleaving GSDMD relative to the WT enzyme (Fig 5G), similar to the human mutants that could only cleave, but not completely remove the IDL (Fig 4). Overall, these data demonstrate that mouse pro-caspase-1 IDL processing, like human IDL processing, is required for GSDMD cleavage and pyroptosis.

Here, we have shown that the human NLRP1 and CARD8 inflammasomes are remarkably distinct. First, these PRRs have functionally divergent C-terminal UPA-CARD fragments—one that

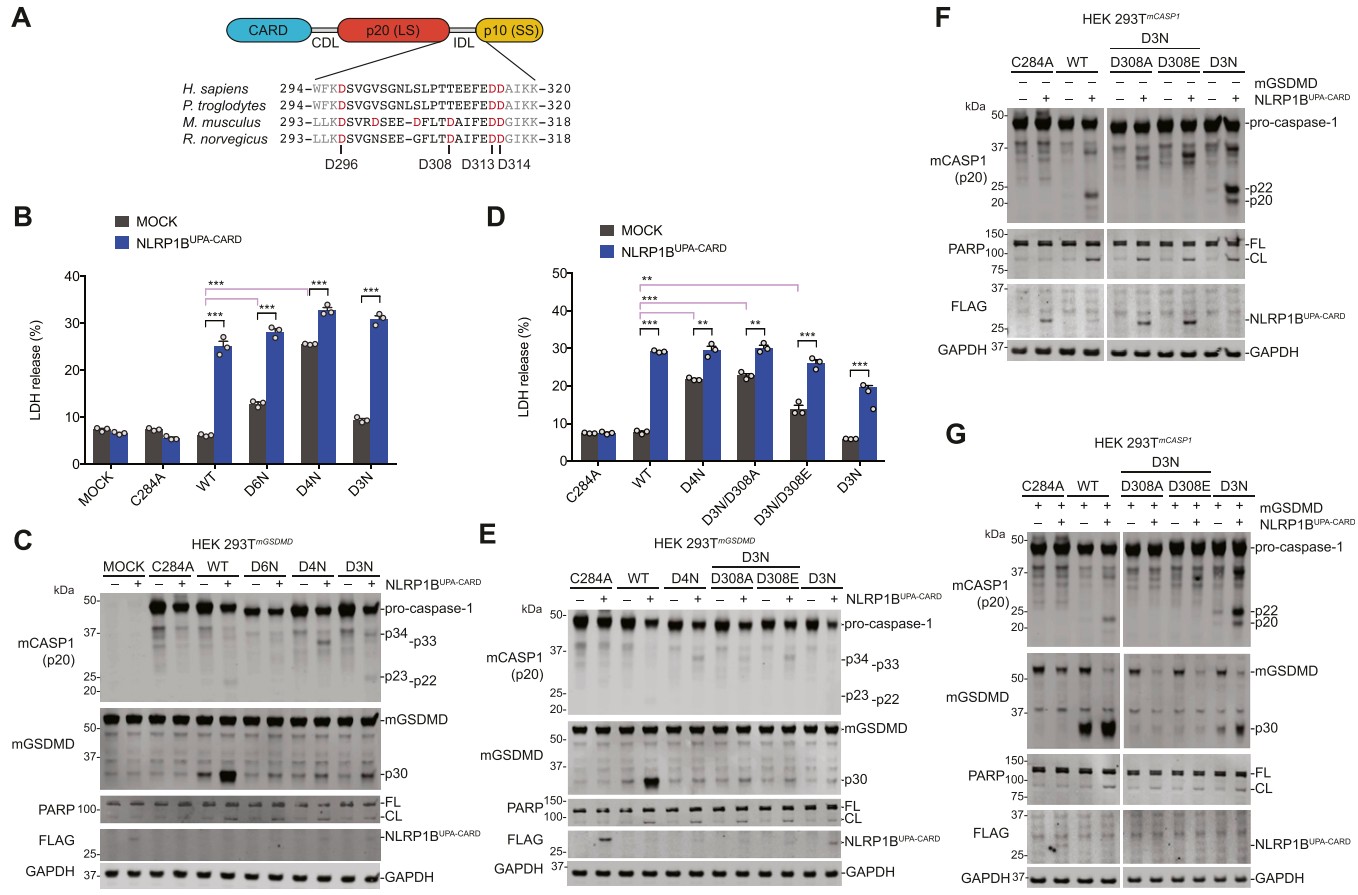

**Figure 5. Cleavage of the mouse pro-caspase-1 interdomain linker is critical for pyroptosis.**
**(A)** Schematic of pro-caspase-1 domain organization and interdomain linker sequences of the indicated species. **(B, C, D, E)** HEK 293T cells stably expressing mouse GSDMD were transiently transfected with plasmids encoding the indicated mouse pro-caspase-1 proteins (0.01 μg) ± NLRP1B[UPA-CARD] (0.02 μg) for 24 h before the collection of supernatants for quantification of LDH release (B, D) and harvesting cell lysates for immunoblots (C, E). **(F, G)** HEK 293T cells stably expressing various mouse pro-caspase-1 constructs were evaluated by immunoblotting after transient transfection of plasmids encoding NLRP1B[UPA-CARD] alone (0.02 μg) (F) or together with mouse GSDMD (0.01 μg) (G) for 24 h. **(F, G)** Cropped images in (F, G) are from the same membrane. Data are mean values ± SEM of three biological replicates. *$P < 0.01$, **$P < 0.01$, ***$P < 0.001$ by two-sided $t$ test. All data, including immunoblots, are representative of three or more independent experiments.

induces an ASC focus to indirectly activate pro-caspase-1 and one that directly activates pro-caspase-1. As such, we predict that the physiological outputs of NLRP1 and CARD8 activation will be different in vivo, for example, in the kinetics of immune activation or in the type or extent of cytokine processing. Future investigations are needed to establish the biological purpose of these ASC-independent and ASC-dependent inflammasomes. Second, CARD8 and NLRP1 have entirely different N-terminal fragments. Although both are activated by at least one similar signal—the cellular consequence of DPP8/9 inhibition—we speculate that these N-terminal fragments likely evolved for different purposes that remain to be elucidated.

More generally, we have now demonstrated that pro-caspase-1 IDL cleavage is necessary for pyroptosis induced by both ASC-dependent and ASC-independent inflammasomes. Interestingly, cleavage at either end of the IDL enables some GSDMD cleavage, but, consistent with previous in vitro data (Elliott et al, 2009), complete removal of the IDL by cleavage at both ends is required for maximal activity. Autoproteolysis is similarly a key step in the activation of other caspases (Kang et al, 2008; Kallenberger et al, 2014). In particular, two recent studies have demonstrated that the

related inflammatory caspase-11, which only forms an ASC-independent inflammasome (termed the "non-canonical" inflammasome) with often little detectable self-cleavage and no direct IL-1β processing (Hagar et al, 2013; Yang et al, 2015), also requires IDL autoproteolysis for activation (Lee et al, 2018; Ross et al, 2018). In this way, the ASC-independent caspase-1 canonical inflammasome is similar to the non-canonical caspase-11 inflammasome. Collectively, these reports and our data show that limited proteolysis within the IDL linker plays a critical role in the activation of inflammatory caspases.

# Materials and Methods

## Antibodies and reagents

Antibodies used include GSDMD rabbit polyclonal Ab (NBP2-33422; Novus Biologicals), human NLRP1/NALP1 sheep polyclonal antibody (AF6788; R&D systems), human NLRP1/NALP1 mouse monoclonal antibody (#447916; R&D systems) V5 rabbit polyclonal Ab (Ab9116;

Abcam), FLAG M2 monoclonal Ab (F3165; Sigma-Aldrich), Myc-Tag (71D10) rabbit monoclonal Ab (#2278; Cell Signaling Technology), CARD8 N terminus rabbit polyclonal antibody (Ab194585; Abcam), CARD8 C terminus rabbit polyclonal Ab (Ab24186; Abcam), human ASC sheep polyclonal antibody (AF3805; R&D systems), GAPDH rabbit monoclonal Ab (14C10; Cell Signaling Technology), NLuc (Lg-BiT) polyclonal antibody (courtesy of Promega), human caspase-1 p20 Rabbit polyclonal Ab (#2225; Cell Signaling Technology), human caspase-1 p10/20 Rabbit polyclonal antibody (16804-1-AP; Proteintech), PARP Rabbit polyclonal Ab (#9542; Cell Signaling Technology), IRDye 680 RD streptavidin (926-68079; LI-COR), IRDye 800CW antirabbit (925-32211; LI-COR), IRDye 800CW antimouse (925-32210; LI-COR), IRDye 680CW antirabbit (925-68073; LI-COR), and IRDye 680CW antimouse (925-68072; LI-COR). Other reagents used include Val-boroPro (VbP) (Okondo et al, 2017), bortezomib (504314; MilliporeSigma), MG132 (474790; MilliporeSigma), carfilzomib (17554; Cayman Chemical), B/B homodimerizer (635059, equivalent to AP-20187; Takara), DSS (21655; Thermo Fisher Scientific), and FuGENE HD (E2311; Promega).

### Cell culture

HEK 293T cells and THP-1 cells were purchased from American Type Culture Collection. HEK 293T cells were grown in DMEM with L-glutamine and 10% FBS. THP-1 cells were grown in Roswell Park Memorial Institute medium 1640 with L-glutamine and 10% FBS. All cells were grown at 37°C in a 5% $CO_2$ atmosphere incubator. Cell lines were regularly tested for mycoplasma using the MycoAlert Mycoplasma Detection Kit (Lonza). Stable cell lines were generated as described previously (Johnson et al, 2018).

### Cloning

Plasmids for full-length and truncated CARD8, NLRP1, mouse NLRP1B (allele 1), mouse and human GSDMD, and mouse and human CASP1 (transcript variant alpha) (Okondo et al, 2017, 2018; Johnson et al, 2018) were cloned as previously described. Unless otherwise stated, all constructs prepared for transient transfection or lentiviral infection of cells were shuttled into modified pLEX_307 vectors (Addgene) using Gateway technology (Thermo Fisher Scientific). A plasmid encoding NLRC4 was purchased from Origene (RC206757) and a construct encoding residues 1–328 was cloned into the Gateway system. A pLEX_307 vector containing mRFP was used for transfection controls. Point mutations were generated using the QuikChange II site-directed mutagenesis kit (200523; Agilent) following the manufacturer's instructions. The NLRP1ΔPYD construct starts from Ser93 of human origin. DNA encoding SmBiT and LgBiT for the NanoBiT assay (Promega) were inserted after the *attR2* recombination site in a modified pLEX_307 vector (immediately after the EcoRV site), and DNA encoding specific CARD domains or full-length ASC were shuttled into these modified vectors using Gateway technology. DmrB-caspase-1 chimera constructs were cloned via assembly PCR reactions using template DNA from caspase-1 or IDL mutants obtained through QuikChange II site-directed mutagenesis. Chimeras encode caspase-1 from D92 onwards.

### Transient transfections

HEK 293T cells were plated in six-well culture plates at $5.0 \times 10^5$ cells/ well in DMEM. The next day, the indicated plasmids were mixed with an empty vector to a total of 2.0 $\mu$g DNA in 125 $\mu$l in Opti-MEM and transfected using FuGENE HD (Promega) according to the manufacturer's protocol. Unless indicated otherwise, 0.02 $\mu$g CARD8, 0.02 $\mu$g NLRP1, and 0.005 $\mu$g ASC were used. The next day, the cells were treated as described. For microscopy experiments, the cells were plated directly into Nunc LabTek II Chamber slide w/Cover sterile glass slides (154534; Thermo Fisher Scientific) at $8.0 \times 10^4$ cells/well and treated with 25 $\mu$l transfection master mix dropwise.

### LDH cytotoxicity and immunoblotting assays

HEK 293T cells were transiently transfected and inhibitor treated as indicated. THP-1 cells were plated in six-well culture plates at $5.0 \times 10^5$ cells/well and treated with VbP as indicated. 15 min before the conclusion of cell transfection experiments, 80 $\mu$l of a 9% Triton X-100 solution was added to designated lysis control wells of a six-well culture plate to completely lyse the cell contents. Supernatants were analyzed for LDH activity using the Pierce LDH Cytotoxicity Assay Kit (Life Technologies). For immunoblotting, the cells were washed 2× in PBS (pH = 7.4), resuspended in PBS, and lysed by sonication. Protein concentrations were determined using the DCA Protein Assay Kit (Bio-Rad). The serum-free supernatants were concentrated using the Amicon Ultra-15 3-kD molecular weight cutoff centrifugal filter unit before immunoblotting. The samples were separated by SDS–PAGE, immunoblotted, and visualized using the Odyssey Imaging System (LI-COR).

### Fluorescence microscopy

Imaging was performed on a Zeiss Axio Observer.Z1 inverted wide-field microscope using 40×/0.95NA air objective. The cells were plated on LabTek eight-well chambered cover glass (155409; Thermo Fisher Scientific) with #1 coverslip. For each chamber, 10 positions were imaged on bright-field, red, and green fluorescence channels at a single time point from a given experiment. Data were exported as raw .czi files and analyzed using custom macro written in ImageJ/FIJI. Total cell area was estimated from RFP-positive signal, and the number of GFP-ASC specks was quantified using the "Analyze particles" function following threshold adjustment in the GFP positive images.

### Split luciferase assay

HEK cells were seeded at $3.0 \times 10^6$ cells in 10-cm dishes and transfected with 3 $\mu$g of the indicated DNA construct using FuGENE HD (Promega). 24 h posttransfection, the cells were washed with cold PBS (Corning), harvested by scraping, and pelleted at 450$g$ for 3 min. The pellets were resuspended in 500 $\mu$l PBS and lysed by sonication. The lysates were clarified to remove bulk cellular debris by centrifugation at 1,000$g$ for 5 min, and relative expression was normalized by gel densitometry of immunoblots (ImageJ 1.52n software). NanoBiT assays were carried out in quadruplicate in white, clear, flat-bottom, 384-well assay plates (3765; Corning).

Equal volume aliquots of the corresponding SmBiT/LgBiT pairs were combined within each well from normalized lysates, followed by addition of Nano-Glo Live Cell Reagent, prepared as per the manufacturer's instructions. After thermal equilibration, luminescence was read on a Cytation 5 multi-modal plate reader.

## DSS cross-linking

HEK 293T cells were treated as indicated before lysates were harvested and pelleted at $400g$ 4°C for 3 min and washed with cold PBS. The cell pellets were lysed with 200 $\mu$l 0.5% NP-40 in TBS for 30 min on ice in 1.75-ml micro-centrifuge tubes. The lysates were spun down at $1,000g$ 4°C for 10 min to remove bulk cell debris (100 $\mu$l of supernatant was reserved for immunoblot). The remaining lysate was placed in the centrifuge for 10 min at $20,000g$ 4°C. The obtained pellet was then washed with 100 $\mu$l CHAPS buffer (50 mM Hepes pH 7.5, 5 mM $MgCl_2$, 0.5 mM EGTA, and 0.1% wt/vol CHAPS) and then resuspended in 48 $\mu$l CHAPS buffer. 2 $\mu$l of 250 mM DSS was added and the samples were agitated at 37°C on an Eppendorf Thermomixer C for 45 min to facilitate protein cross-linking. The samples were then combined with an equal volume of 2× loading dye and heated to 98°C for 10 min and prepared for immunoblot analysis.

## Generation of stable cell lines

For generating HEK 293T and THP-1 cells ectopically expressing pro-caspase-1 constructs, the constructs were packaged into lentivirus in HEK 293T cells using the Fugene HD transfection reagent (Promega) and 2 $\mu$g of the vector, 2 $\mu$g psPAX2, and 1 $\mu$g pMD2.G. The virus was concentrated using the PEG Virus Precipitation Kit (ab102538; Abcam), and $CASP1^{-/-}$ THP-1 cells made with sgRNA1 from our previous report (Okondo et al, 2017) were spinfected with virus for 1 h at $1,000g$ at 30°C supplemented 10 $\mu$g/ml polybrene. For HEK 293T cells, the virus was not concentrated. After 2 d, the cells were selected for stable expression of CASP1 using hygromycin (100 $\mu$g/ml).

## Data analysis and statistics

Statistical analysis was performed using GraphPad Prism 7.0 software. Statistical significance was determined using two-sided $t$ tests.

## Significance

We demonstrate that the related human NLRP1 and CARD8 exclusively form ASC-dependent and ASC-independent inflammasomes, respectively. In contrast to the current model for caspase-1 activation, we show that both types of inflammasomes require pro-caspase-1 processing within the IDL for the induction of pyroptosis.

## Supplementary Information

## Acknowledgements

We thank K Schroder and P Broz for sharing DmrB constructs. This work was supported by the Josie Robertson Foundation (DA Bachovchin); a Stand Up to Cancer-Innovative Research Grant (Grant Number SU2C-AACR-IRG11-17 to DA Bachovchin; Stand Up to Cancer is a program of the Entertainment Industry Foundation, and Research Grants are administered by the American Association for Cancer Research, the scientific partner of SU2C); the Pew Charitable Trusts (DA Bachovchin is a Pew-Stewart Scholar in Cancer Research); the Pershing Square Sohn Cancer Research Alliance (DA Bachovchin); the National Institutes of Health (NIH) (R01 AI137168 to DA Bachovchin, T32 GM007739-Andersen to AR Griswold, NIH T32 GM115327-Tan to EL Orth and LE Vostal, the Memorial Sloan Kettering Cancer Center Core Grant P30 CA008748, and F30 CA243444 to AR Griswold); an Alfred P Sloan Foundation Research Fellowship (DA Bachovchin); Gabrielle's Angel Foundation (DA Bachovchin); the American Cancer Society (Postdoctoral Fellowship PF-17-224-01 – CCG to CY Taabazuing); Mr William H and Mrs Alice Goodwin, the Commonwealth Foundation for Cancer Research, and The Center for Experimental Therapeutics of Memorial Sloan Kettering Cancer Center (DA Bachovchin); and The Ludwig Center at Memorial Sloan Kettering Cancer Center (DA Bachovchin).

## Author Contributions

DP Ball: conceptualization, data curation, formal analysis, funding acquisition, investigation, methodology, and writing—original draft, review, and editing.
CY Taabazuing: conceptualization, data curation, formal analysis, funding acquisition, investigation, methodology, and writing—original draft, review, and editing.
AR Griswold: data curation, formal analysis, investigation, and writing—review and editing.
EL Orth: formal analysis, investigation, and writing—review and editing.
SD Rao: investigation and writing—review and editing.
IB Kotliar: formal analysis and investigation.
LE Vostal: formal analysis and investigation.
DC Johnson: investigation.
DA Bachovchin: conceptualization, formal analysis, supervision, funding acquisition, and writing—original draft, review, and editing.

## Conflict of Interest Statement

The authors declare that they have no conflict of interest.

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
