## [Reviewer comments · Life Science Alliance]

Life Science Alliance

Caspase-1 interdomain linker cleavage is required for pyroptosis

Daniel Ball, Cornelius Taabazuing, Andrew Griswold, Elizabeth Orth, Sahana Rao, Ilana Kotliar, Lauren Vostal, Darren Johnson, and Daniel Bachovchin

DOI: <https://doi.org/10.26508/lsa.202000664>

Corresponding author(s): Daniel Bachovchin, Memorial Sloan Kettering Cancer Center

Review Timeline:	Submission Date:	2020-01-31
	Editorial Decision:	2020-02-03
	Revision Received:	2020-02-03
	Accepted:	2020-02-04

Scientific Editor: Andrea Leibfried

Transaction Report:

Please note that the manuscript was previously reviewed at another journal and the reports were taken into account in the decision-making process at Life Science Alliance. Since the original reviews are not subject to Life Science Alliance's transparent review process policy, the reports and author response cannot be published.

February 3, 2020

RE: Life Science Alliance Manuscript #LSA-2020-00664-T

Prof. Daniel A Bachovchin
Memorial Sloan Kettering Cancer Center
Chemical Biology Program
1275 York Avenue
Box 428
New York, New York NY 10065

Dear Dr. Bachovchin,

Thank you for transferring your revised manuscript entitled "Caspase-1 interdomain linker cleavage is required for pyroptosis". Your manuscript was reviewed twice at another journal before, and the editors provided those reviewer reports to us with your permission.

While one reviewer was supportive of publication, two of the reviewers who re-evaluated your study at the other venue would have expected a further reaching conceptual advance and more insight in a physiologically-relevant system. We had indicated to you prior to submission, that such physiologically-relevant insight is not needed for publication here. We would thus be happy to publish your paper in Life Science Alliance pending final minor revisions:

- Please provide a full point-by-point response to the remaining concerns. Please address the remaining concerns of reviewer #1 with text changes. Please address the remaining concerns of reviewer #2 in the following way: if possible, please provide evidence for the GSDMD-p30 band in your blots being indeed p30. Please check carefully throughout your manuscript and add information on replicates. Please mention within the manuscript text or figure legend that cells were not fixed prior to ASC spec analysis and change the quantification to % of cells with ASC specks. Please fix the wording in the abstract to respond to the reviewer concern #4 and address the minor issues.
- Please add a summary blurb and author contributions (to be selected for each author in the author area when submitting) within our submission system
- Please provide all figures, including supplementary figures, as individual files, the legends should remain in the main manuscript file (in a separate section)
- Please provide the main ms file in word docx format

A. FINAL FILES:

B. MANUSCRIPT ORGANIZATION AND FORMATTING:

Thank you for your attention to these final processing requirements.

Sincerely,

LSA Point-by-point response

While one reviewer was supportive of publication, two of the reviewers who re-evaluated your study at the other venue would have expected a further reaching conceptual advance and more insight in a physiologically-relevant system. We had indicated to you prior to submission, that such physiologically-relevant insight is not needed for publication here. We would thus be happy to publish your paper in Life Science Alliance pending final minor revisions:

- Please provide a full point-by-point response to the remaining concerns.

Our responses are in blue

Please address the remaining concerns of reviewer #1 with text changes.

We have amended the text to include the specific items requested, including the sgRNA used for the *CASP1* KO THP-1 cell line and the sequence information of human caspase-1. The identity of the Lentiguide puro vector was already listed in the "cloning" methods section.

Please address the remaining concerns of reviewer #2 in the following way: if possible, please provide evidence for the GSDMD-p30 band in your blots being indeed p30.

The labeled GSDMD-p30 band is indeed p30. VbP is well-characterized to induce the formation of this GSDMD p30 in WT THP-1 cells. These cells were used in this experiment, and this band is clearly visible in the second lane of the gel, which is precisely the correct size. Thus, this is the correct p30 band.

Please check carefully throughout your manuscript and add information on replicates.

We have carefully checked and added information of replicates throughout the manuscript.

Please mention within the manuscript text or figure legend that cells were not fixed prior to ASC spec analysis and change the quantification to % of cells with ASC specks.

We have now noted in the legend that the cells were not fixed prior to speck analysis. Due to the experimental setup, we cannot quantify the % of cells with ASC specks from these images. However, we should note that not all publications quantify % of cells with specks. For example, Boucher et al (JEM, 2018) report the relative intensity of specks per field of view (Fig. 4C, 5A), Van Opdenbosch et al (*Nat. Comm.* 2014) report #specks/field (Fig. 4B, 4D), and Wickliffe et al (Wickliffe, *Nature*. 2019) quantitate #specks/region of interest (Fig. 4B). Our analysis is in line with these previous analyses. Moreover, we have corroborated our data with gel-based oligomer assays.

Please fix the wording in the abstract to respond to the reviewer concern #4 and address the minor issues.

We have adjusted this wording.

- Please add a summary blurb and author contributions (to be selected for each author in the author area when submitting) within our submission system

Done

- Please provide all figures, including supplementary figures, as individual files, the legends should remain in the main manuscript file (in a separate section)

Done

- Please provide the main ms file in word docx format

Done

February 4, 2020

RE: Life Science Alliance Manuscript #LSA-2020-00664-TR

Prof. Daniel A Bachovchin
Memorial Sloan Kettering Cancer Center
Chemical Biology Program
1275 York Avenue
Box 428
New York, New York NY 10065

Dear Dr. Bachovchin,

Thank you for submitting your Research Article entitled "Caspase-1 interdomain linker cleavage is required for pyroptosis". It is a pleasure to let you know that your manuscript is now accepted for publication in Life Science Alliance. Congratulations on this interesting work.

DISTRIBUTION OF MATERIALS:

Again, congratulations on a very nice paper. I hope you found the review process to be constructive and are pleased with how the manuscript was handled editorially. We look forward to future exciting submissions from your lab.

Sincerely,
